# Application of Laplace–Adomian Decomposition Method for the Analytical Solution of Third-Order Dispersive Fractional Partial Differential Equations

**DOI:** 10.3390/e21040335

**Published:** 2019-03-28

**Authors:** Rasool Shah, Hassan Khan, Muhammad Arif, Poom Kumam

**Affiliations:** 1Department of Mathematics, Abdul Wali khan University, Mardan 23200, Pakistan; 2Center of Excellence in Theoretical and Computational Science (TaCS-CoE) & KMUTTFixed Point Research Laboratory, Room SCL 802 Fixed Point Laboratory, Science Laboratory Building, Departments of Mathematics, Faculty of Science, King Mongkut’s University of Technology Thonburi (KMUTT), 126 Pracha-Uthit Road, Bang Mod, Thrung Khru, Bangkok 10140, Thailand; 3Department of Medical Research, China Medical University Hospital, China Medical University, Taichung 40402, Taiwan

**Keywords:** Laplace–Adomian decomposition method, third-order dispersive equations, Caputo operator, analytical solution

## Abstract

In the present article, we related the analytical solution of the fractional-order dispersive partial differential equations, using the Laplace–Adomian decomposition method. The Caputo operator is used to define the derivative of fractional-order. Laplace–Adomian decomposition method solutions for both fractional and integer orders are obtained in series form, showing higher convergence of the proposed method. Illustrative examples are considered to confirm the validity of the present method. The fractional order solutions that are convergent to integer order solutions are also investigated.

## 1. Introduction

Partial differential equations (PDEs) are used to model different physical phenomena in various areas of applied sciences such as fluid dynamics, mathematical biology, quantum mechanics, chemical kinetics and linear optics. In 1895, the Korteweg–de and Vries derived a non dimensionalized version of an equation, known as the KdV equation. This model is used for the study of dispersive wave phenomena in numerous fields of science and technology, like plasma physics and quantum mechanics. The exact solution of the KdV may not be available, therefore a lot of analytical techniques have been discussed for its approximate solution [1]. There are two important dispersive terms, namely third and fifty order in KdV equations. The KdV equation of order five has been used to describe plasma physics [2]. Numerical solutions of the fifth and third order dispersive KdV equations have been studied in [3].

The non-linear nature is responsible for the complete study of any physical system, which shows the importance of a non-linear term present in any model of the physical problems. In this connection, reductive perturbation theories have been studied in [4] for the non-linear KdV. A variational method has been suggested in [5] for the exact solution of KdV with higher-order nonlinearity. Numerical solutions for the KdV Burgers equation have been successfully derived in [6] by using a compact-type constrained interpolation profile method. Numerical results of KdV equations of order five are presented in [1] using homotopy perturbation transform method. The KdV equations of order three and five have been discussed in [3] by using two numerical methods. Entropies based on fractional calculus [7], integer and fractional dynamical systems can be solved by entropy analysis [8], nonlinear partial differential equations [9] in entropy and convexity, as well as ractional derivative advection–diffusion in two-dimensional semi-conductor systems and the dynamics of a national soccer league [10,11]. The exact solution to differential equations (DEs) of fractional order with mixed partial derivatives [12] are fractional linear differential equations with constant coefficients in Laplace transform [13]. Laplace homotopy analysis method (LHAM) can be used to solve FDEs [14] systems of non-linear FPDEs in a new analytical technique [15]. The solution to non-linear coupled space-time fractional modified KdV equations was obtained by Feng’s first integral method [16]. Fractional PDEs of order three have been solved by different methods, such as fractional variational iteration method (FVIM) [17], classical Riccati equations method [18], fractional differential transform method(FDTM) and modified fractional differential transform method (MFDTM) [19], spline method (SM) [20] and homotopy analysis Sumudu transform method (HASTM) [21].

Among all these methods, the Laplace–Adomian decomposition method (LADM) is an efficient analytical method to solve non-linear FDES. LADM is the combination of two powerful techniques, the Adomian decomposition method and the Laplace transform. Further, LADM does not requireme the predefined size declaration like the Runge–Kutta method. Therefor this technique is considered, an ideal for those equations that represent nonlinear models. As compared to other analytical techniques, LADM have less parameters, therefore LADM is an efficient technique, requiring no discretization and linearization [22]. A comparison between the LADM and ADM for the analysis of FDEs is given in [23]. The Kundu–Eckhaus equation deals with quantum field theory, and the analytical solution of this nonlinear PDEs has been studied in [24] using LADM. Multi-step Laplace Adomian decomposition method have been described in [25] for non-linear FDEs. Analysis of smoke model have been studied successfully by using LADM [26].

In view of the above literature, we made a successful attempt to obtain the analytical results of dispersive FPDEs, using LADM. The results of the LADM are interesting and in better contact with exact solutions for the problems.

## 2. Definitions and Preliminaries of Concepts

In this unit, among a few definitions of fractional calculus, presented in the article due to Riemann Liouville, Grunewald Letnikov, Caputo, etc., first folk’s simple descriptions and introductions are reconsidered, which we want to comprehend for our education.

**Definition** **1.***R–L fractional integral*$$I_{x}^{\gamma }g\left(x\right)=\left\{\begin{array}{cc} g(x) & \mathrm{if}\;\gamma =0\\ \frac{1}{\Gamma (\gamma )}\int_{0}^{x}{\left(x-\upsilon \right)}^{\gamma -1}g\left(\upsilon \right)d\upsilon & \mathrm{if}\;\gamma >0, \end{array}\right.$$*where* Γ *denotes the gamma function define by,*
$$\Gamma \left(\omega \right)=\int_{0}^{\infty }e^{-x}x^{\omega -1}dx\;\omega \in C,$$
*In this study, Caputo et al. [27] suggested a revise fractional derivative operator in order to overcome inconsistency measured in the Riemann Liouville derivative [27,28]. The above mathematical statement described a Caputo fractional derivative operator of initial and boundary condition for fractional as well as integer order derivative.*


**Definition** **2.**
*The Caputo operator of order γ for a fractional derivative is given by the following mathematical expression for n∈N, x>0, $g\in C_{t}$, t≥-1.*
$$D^{\gamma }g\left(x\right)=\frac{\partial^{\gamma }g\left(x\right)}{\partial t^{\gamma }}=\left\{\begin{array}{cc} I^{n-\gamma }\left[\frac{\partial^{\gamma }g\left(x\right)}{\partial t^{\gamma }}\right], & \mathrm{if}\;n-1<\gamma \leq n,\;n\in N\\ \frac{\partial^{\gamma }g\left(x\right)}{\partial t^{\gamma }}, \end{array}\right.$$
*Hence, we require the subsequent properties given in the next Lemma.*


**Lemma** **1.**
*If n-1<γ≤n with n∈N and $g\in C_{t}$ with t≥-1, then*
$$\begin{array}{c} I^{\gamma }I^{a}g\left(x\right)=I^{\gamma +a}g\left(x\right),\;a,\gamma \geq 0.\\ I^{\gamma }x^{\lambda }=\frac{\Gamma (\lambda +1)}{\Gamma (\gamma +\lambda +1)}x^{\gamma +\lambda },\;\gamma >0,\lambda >-1,\;x>0.\\ I^{\gamma }D^{\gamma }g\left(x\right)=g\left(x\right)-\sum_{k=0}^{n-1}g^{\left(k\right)}\left(0^{+}\right)\frac{x^{k}}{k!},\;\mathrm{for}\;x>0,n-1<\gamma \leq n. \end{array}$$
*In current study, the Caputo operator is reasonable as other fractional derivative operators have certain disadvantages. Further information about fractional derivatives, are found in [29,30].*


**Definition** **3.**
*The Laplace transform of h(t),t>0 is defined by [31]*
$$H(s)=L\left[h(t)\right]=\int_{0}^{\infty }e^{-st}h\left(t\right)dt.$$


**Definition** **4.**
*The Laplace transform in term of convolution is given by*
$$L\left[h_{1}*h_{2}\right]=L\left[h_{1}\left(t\right)\right]*L\left[h_{2}\left(t\right)\right],$$
*here $h_{1}*h_{2}$ define the convolution between $h_{1}$ and $h_{2}$,*
$$\left(h_{1}*h_{2}\right)t=\int_{0}^{\tau }h_{1}\left(\tau \right)h_{2}\left(t-\tau \right)dt.$$

*The fractional derivative in term of a Laplace transform is*
$$L\left(D_{t}^{\gamma }h\left(t\right)\right)=s^{\gamma }H\left(s\right)-\sum_{k=0}^{n-1}s^{\gamma -1-k}h^{\left(k\right)}\left(0\right),\;n-1<\gamma <n,$$
*where H(s) is the Laplace transform of h(t).*


**Definition** **5.**
*The Mittag–Leffler function, $E_{\gamma }\left(p\right)$ for γ>0 is represented as,*
$$E_{\gamma }\left(p\right)=\sum_{n=0}^{\infty }\frac{p^{n}}{\Gamma (\gamma n+1)}\;\gamma >0,\;p\in C.$$


**Theorem** **1.**
*Here, we will study the convergence analysis as same manner in [32] of the LADM applied to the fractional dispersive PDE of order three. Let us consider the Hilbert space H which may define by $H=L^{2}\left(\left(\alpha ,\beta \right)X\left[0,T\right]\right)$ the set of applications:*
$$u:\left(\alpha ,\beta \right)X\left[0,T\right]\to \;with\;\int_{\left(\alpha ,\beta )X[0,T\right]}u^{2}\left(x,s\right)dsd\theta \;<+\infty .$$

*Now we consider the fractional dispersive PDE of order three in the above assumptions and let us denote*
$$L\left(u\right)=\frac{\partial^{\gamma }u}{\partial t^{\gamma }},$$
*then the fractional dispersive PDE becomes, in an operator form*
$$L\left(u\right)=-\varphi \frac{\partial \nu (x,t)}{\partial x}-w\frac{\partial^{3}\nu \left(x,t\right)}{\partial x^{3}}.$$

*The LADM is convergence, if the following two hypotheses are satisfied:*
$$\left(H1\right)\left(L\left(u\right)-L\left(v\right),u-v\right)\geq k∥u-v{∥}^{2};k>0,\forall u,v\epsilon H.$$

*H(2) whatever may be M>0, there exist a constant C(M)>0 such that for u,vϵH with ∥u∥≤M, ∥v∥≤M we have (L(u)-L(v),u-v)≤C(M)∥u-v∥∥w∥ for every wϵH.*


## 3. Idea of Fractional Laplace–Adomian Decomposition Method

### 3.1. LADM for Dispersive Equation of One-Dimensional

In this section, LADM is implemented to solve fractional dispersive PDE of order three.
(1)$$\frac{\partial^{\gamma }\nu \left(x,t\right)}{\partial t^{\gamma }}+w\frac{\partial^{3}\nu \left(x,t\right)}{\partial x^{3}}=q\left(x,t\right),\;w,t\geq 0,\;m-1<\gamma <m,$$
q(x,t) is the source function.

Subject to initial and boundary conditions
(2)ν(x,0)=k(x),
(3)$$\begin{array}{c} \nu \left(0,t\right)=l_{0}\left(t\right),\\ \nu_{x}\left(0,t\right)=l_{1}\left(t\right),\\ \nu_{xx}\left(0,t\right)=l_{2}\left(t\right). \end{array}$$

With the help of Laplace transform, Equation (Equation 1) can be written as
(4)$$L\left[\frac{\partial^{\gamma }\nu \left(x,t\right)}{\partial t^{\gamma }}\right]+L\left[w\frac{\partial^{3}\nu \left(x,t\right)}{\partial x^{3}}\right]=L\left[q(x,t)\right],$$
(5)$$L\left[\nu (x,t)\right]=\frac{k(x)}{s}+\frac{1}{s^{\gamma }}L\left[q(x,t)\right]-\frac{1}{s^{\gamma }}L\left[w\frac{\partial^{3}\nu \left(x,t\right)}{\partial x^{3}}\right].$$

The LADM solution ν(x,t) is represented by the following infinite series
(6)$$\nu \left(x,t\right)=\sum_{j=0}^{\infty }\nu_{j}\left(x,t\right),$$
and the nonlinear terms (if any) in problem are defined by the infinite series of Adomian polynomials,
(7)$$N\nu (x,t)=\sum_{j=0}^{\infty }A_{j},$$
(8)$$A_{j}=\frac{1}{j!}{\left[\frac{d^{j}}{d\lambda^{j}}\left[N\sum_{j=0}^{\infty }\left(\lambda^{j}\nu_{j}\right)\right]\right]}_{\lambda =0},\;j=0,1,2...\;,$$
and substitution of Equations (Equation 5) and (Equation 6) into Equation (Equation 4), we get
(9)$$L\left[\sum_{j=0}^{\infty }\nu \left(x,t\right)\right]=\frac{k(x)}{s}+\frac{1}{s^{\gamma }}L\left[q(x,t)\right]-\frac{1}{s^{\gamma }}L\left[w\frac{\partial^{3}\nu_{j}\left(x,t\right)}{\partial x^{3}}\right].$$

Applying the linearity of the Laplace transform,
$$L\left[\nu_{0}\left(x,t\right)\right]=\frac{\nu (x,0)}{s}+\frac{1}{s^{\gamma }}L\left[q(x,t)\right]=k\left(x,s\right),$$
$$L\left[\nu_{1}\left(x,t\right)\right]=-\frac{1}{s^{\gamma }}L\left[w\frac{\partial^{3}\nu_{0}\left(x,t\right)}{\partial x^{3}}\right].$$

Generally, we can write
(10)$$L\left[\nu_{j+1}\left(x,t\right)\right]=-\frac{1}{s^{\gamma }}L\left[\frac{\partial^{3}\nu_{j}\left(x,t\right)}{\partial x^{3}}\right],\;j\geq 1.$$

Applying the inverse Laplace transform in Equation (Equation 9)
$$\nu_{0}\left(x,t\right)=k\left(x,t\right),$$
(11)$$\nu_{j+1}\left(x,t\right)=-L^{-1}\left[\frac{1}{s^{\gamma }}L\left[\frac{\partial^{3}\nu_{j}\left(x,t\right)}{\partial x^{3}}\right]\right].$$

### 3.2. LADM for Dispersive Equation of Higher-Dimension

A dispersive PDE in higher dimension is represented as,
(12)$$\begin{array}{c} \nu_{t}^{\gamma }+c\nu_{xxx}+d\nu_{yyy}+e\nu_{zzz}=q\left(x,y,z,t\right),\\ \;t\geq 0,\;c,d,e\geq 0,\;m-1<\gamma <m, \end{array}$$
where the source function is denoted by q(x,y,z,t). The initial condition is
(13)ν(x,y,z,0)=k(x,y,z).

With the help of a Laplace transform, Equation (Equation 12) can be written as
(14)$$L\left[\nu_{t}^{\gamma }\right]+L\left[c\nu_{xxx}+d\nu_{yyy}+e\nu_{zzz}\right]=L\left[q(x,y,z,t)\right],$$
and using the differentiation property of the Laplace transform, we get
(15)$$\begin{array}{c} L\left[\sum_{j=0}^{\infty }\nu \left(x,y,z,t\right)\right]=\frac{k(x,y,z)}{s}+\frac{1}{s^{\gamma }}L\left[q(x,y,z,t)\right]\\ -\frac{1}{s^{\gamma }}L\left[c\nu_{xxx}+d\nu_{yyy}+e\nu_{zzz}\right]. \end{array}$$

Applying the linearity of the Laplace transform,
$$L\left[\nu_{0}\left(x,y,z,t\right)\right]=\frac{\nu (x,y,z,0)}{s}+\frac{1}{s^{\gamma }}L\left[q(x,y,z,t)\right]=k\left(x,y,z,s\right),$$
$$L\left[\nu_{1}\left(x,y,z,t\right)\right]=-\frac{1}{s^{\gamma }}L\left[c\nu_{0xxx}+d\nu_{0yyy}+e\nu_{0zzz}\right].$$

Generally, we can write
(16)$$L\left[\nu_{j+1}\left(x,y,z,t\right)\right]=-\frac{1}{s^{\gamma }}L\left[c\nu_{jxxx}+d\nu_{jyyy}+e\nu_{jzzz}\right],\;j\geq 1.$$

Applying the inverse Laplace transform, in Equation (Equation 15)
$$\nu_{0}\left(x,y,z,t\right)=k\left(x,y,z,t\right),$$
(17)$$\nu_{j+1}\left(x,y,z,t\right)=-L^{-1}\left[\frac{1}{s^{\gamma }}L\left[c\nu_{jxxx}+d\nu_{jyyy}+e\nu_{jzzz}\right]\right].$$

## 4. Results

**Example** **1.**
*Consider the following fractional dispersive KdV in Equation [33]*
(18)$$\frac{\partial^{\gamma }\nu }{\partial t^{\gamma }}+2\frac{\partial \nu }{\partial x}+\frac{\partial^{3}\nu }{\partial x^{3}}=0,\;t>0,\;0<\gamma \leq 1,$$
*with the initial condition*
(19)ν(x,0)=sinx.


Taking the Laplace transform of Equation (Equation 18),
$$L\left[\frac{\partial^{\gamma }\nu }{\partial t^{\gamma }}\right]=-L\left[2\frac{\partial \nu }{\partial x}+\frac{\partial^{3}\nu }{\partial x^{3}}\right],$$
$$s^{\gamma }L\left[\nu (x,t)\right]-s^{\gamma -1}\left[\nu (x,0)\right]=-L\left[2\frac{\partial \nu }{\partial x}+\frac{\partial^{3}\nu }{\partial x^{3}}\right].$$

Applying the inverse Laplace transform
$$\nu (x,t)=L^{-1}\left[\frac{\nu (x,0)}{s}-\frac{1}{s^{\gamma }}L\left[2\frac{\partial \nu }{\partial x}+\frac{\partial^{3}\nu }{\partial x^{3}}\right]\right],$$
$$\nu (x,t)=\sin x-L^{-1}\left[\frac{1}{s^{\gamma }}L\left[2\frac{\partial \nu }{\partial x}+\frac{\partial^{3}\nu }{\partial x^{3}}\right]\right].$$

Using the ADM procedure, we get
$$\sum_{j=0}^{\infty }\nu_{j}\left(x,t\right)=\sin x-L^{-1}\left[\frac{1}{s^{\gamma }}L\left[2\sum_{j=0}^{\infty }\frac{\partial \nu_{j}}{\partial x}+\sum_{j=0}^{\infty }\frac{\partial^{3}\nu_{j}}{\partial x^{3}}\right]\right],$$
(20)$$\begin{array}{c} \nu_{0}\left(x,t\right)==\sin x, \end{array}$$
$$\nu_{j+1}\left(x,t\right)=-L^{-1}\left[\frac{1}{s^{\gamma }}L\left[2\sum_{j=0}^{\infty }\frac{\partial \nu_{j}}{\partial x}+\sum_{j=0}^{\infty }\frac{\partial^{3}\nu_{j}}{\partial x^{3}}\right]\right],$$
for j=0,1,2,..
(21)$$\begin{array}{c} \nu_{1}\left(x,t\right)=-L^{-1}\left[\frac{1}{s^{\gamma }}L\left[2\frac{\partial \nu_{0}}{\partial x}+\frac{\partial^{3}\nu_{0}}{\partial x^{3}}\right]\right],\\ \nu_{1}\left(x,t\right)=-L^{-1}\left[\frac{\cos x}{s^{\gamma +1}}\right]=-\cos x\frac{t^{\gamma }}{\Gamma (\gamma +1)},\\ \nu_{2}\left(x,t\right)=-L^{-1}\left[\frac{1}{s^{\gamma }}L\left[2\frac{\partial \nu_{1}}{\partial x}+\frac{\partial^{3}\nu_{1}}{\partial x^{3}}\right]\right],\\ \nu_{2}\left(x,t\right)=-L^{-1}\left[\frac{\sin x}{s^{2\gamma +1}}\right]=-\sin x\frac{t^{2\gamma }}{\Gamma (2\gamma +1)}. \end{array}$$

The subsequent terms are
(22)$$\begin{array}{c} \nu_{3}\left(x,t\right)=-L^{-1}\left[\frac{1}{s^{\gamma }}L\left[2\frac{\partial \nu_{2}}{\partial x}+\frac{\partial^{3}\nu_{2}}{\partial x^{3}}\right]\right],\\ \nu_{3}\left(x,t\right)=-L^{-1}\left[\frac{\cos x}{s^{3\gamma +1}}\right]=\cos x\frac{t^{3\gamma }}{\Gamma (3\gamma +1)}. \end{array}$$

The LADM solution for Example 1 is
$$\nu \left(x,t\right)=\nu_{0}\left(x,t\right)+\nu_{1}\left(x,t\right)+\nu_{2}\left(x,t\right)+\nu_{3}\left(x,t\right)+...,$$
$$\begin{array}{c} \nu \left(x,t\right)=\sin x-\cos x\frac{t^{\gamma }}{\Gamma (\gamma +1)}-\sin x\frac{t^{2\gamma }}{\Gamma (2\gamma +1)}+\cos x\frac{t^{3\gamma }}{\Gamma (3\gamma +1)}+..., \end{array}$$
and so on. The solution in a series form is given by
(23)$$\begin{array}{c} \nu \left(x,t\right)=\sin x\left(1-\frac{t^{2\gamma }}{\Gamma (2\gamma +1)}+\frac{t^{4\gamma }}{\Gamma (4\gamma +1)}-...\right)\\ -\cos x\left(\frac{t^{\gamma }}{\Gamma (\gamma +1)}-\frac{t^{3\gamma }}{\Gamma (3\gamma +1)}+\frac{t^{5\gamma }}{\Gamma (5\gamma +1)}-...\right), \end{array}$$
when γ=1, then the LADM solution is
(24)$$\begin{array}{c} \nu (x,t)=\sin\left(x-t\right). \end{array}$$

Figure 1 consists of four graphs; (a) the exact solution of ν(x,t) and (b) LADM solution ν(x,t) of Example 1 at γ=1. Figure 1a,b indicate that the present method has strong agreement with the exact solution for the problem. In Figure 1c,d, two graphs (c) and (d) are given, that represent the analytical solution of Example 1 at fractional γ=0.75 and 0.50, respectively. Figure 1c,d reflects that fractional-order approaches to integer order solution surfaces of fractional order are convergent to the integer order surface. It means that physically we can model any of the surfaces as desired by the physical phenomena occurring in nature.

**Example** **2.**
*We next consider the following fractional dispersive KdV equation [33]*
(25)$$\frac{\partial^{\gamma }\nu }{\partial t^{\gamma }}+\frac{\partial^{3}\nu }{\partial x^{3}}+\frac{\partial^{3}\nu }{\partial y^{3}}=0,\;t>0,\;0<\gamma \leq 1,$$
*with initial condition*
(26)ν(x,y,0)=cos(x+y).


Taking Laplace transform of Equation (Equation 25),
$$L\left[\frac{\partial^{\gamma }\nu }{\partial t^{\gamma }}\right]=-L\left[\frac{\partial^{3}\nu }{\partial x^{3}}+\frac{\partial^{3}\nu }{\partial y^{3}}\right],$$
$$s^{\gamma }L\left[\nu (x,y,t)\right]-s^{\gamma -1}\left[\nu (x,y,0)\right]=-L\left[\frac{\partial^{3}\nu }{\partial x^{3}}+\frac{\partial^{3}\nu }{\partial y^{3}}\right].$$

Applying the inverse Laplace transform
$$\nu (x,y,t)=L^{-1}\left[\frac{\nu (x,y,0)}{s}-\frac{1}{s^{\gamma }}L\left[\frac{\partial^{3}\nu }{\partial x^{3}}+\frac{\partial^{3}\nu }{\partial y^{3}}\right]\right],$$
$$\nu (x,y,t)=\cos\left(x+y\right)-L^{-1}\left[\frac{1}{s^{\gamma }}L\left[\frac{\partial^{3}\nu }{\partial x^{3}}+\frac{\partial^{3}\nu }{\partial y^{3}}\right]\right].$$

Using ADM procedure, we get
$$\sum_{j=0}^{\infty }\nu_{j}\left(x,y,t\right)=\cos\left(x+y\right)-L^{-1}\left[\frac{1}{s^{\gamma }}L\left[\sum_{j=0}^{\infty }\frac{\partial^{3}\nu_{j}}{\partial x^{3}}+\sum_{j=0}^{\infty }\frac{\partial^{3}\nu_{j}}{\partial y^{3}}\right]\right],$$
(27)$$\begin{array}{c} \nu_{0}\left(x,y,t\right)=\cos\left(x+y\right), \end{array}$$
$$\nu_{j+1}\left(x,y,t\right)=-L^{-1}\left[\frac{1}{s^{\gamma }}L\left[\sum_{j=0}^{\infty }\frac{\partial^{3}\nu_{j}}{\partial x^{3}}+\sum_{j=0}^{\infty }\frac{\partial^{3}\nu_{j}}{\partial y^{3}}\right]\right],$$
for j=0,1,2,..
(28)$$\begin{array}{c} \nu_{1}\left(x,y,t\right)=-L^{-1}\left[\frac{1}{s^{\gamma }}L\left[\frac{\partial^{3}\nu_{0}}{\partial x^{3}}+\frac{\partial^{3}\nu_{0}}{\partial y^{3}}\right]\right],\\ \nu_{1}\left(x,y,t\right)=-2\sin\left(x+y\right)L^{-1}\left[\frac{1}{s^{\gamma +1}}\right]=-2\sin\left(x+y\right)\frac{t^{\gamma }}{\Gamma (\gamma +1)},\\ \nu_{2}\left(x,y,t\right)=-L^{-1}\left[\frac{1}{s^{\gamma }}L\left[\frac{\partial^{3}\nu_{1}}{\partial x^{3}}+\frac{\partial^{3}\nu_{1}}{\partial y^{3}}\right]\right],\\ \nu_{2}\left(x,y,t\right)=-4\cos\left(x+y\right)L^{-1}\left[\frac{1}{s^{2\gamma +1}}\right]=-4\cos\left(x+y\right)\frac{t^{2\gamma }}{\Gamma (2\gamma +1)}. \end{array}$$

The subsequent terms are
(29)$$\begin{array}{c} \nu_{3}\left(x,y,t\right)=-L^{-1}\left[\frac{1}{s^{\gamma }}L\left[\frac{\partial^{3}\nu_{2}}{\partial^{3}x}+\frac{\partial^{3}\nu_{2}}{\partial y^{3}}\right]\right],\\ \nu_{3}\left(x,y,t\right)=8\sin\left(x+y\right)L^{-1}\left[\frac{1}{s^{3\gamma +1}}\right]=8\sin\left(x+y\right)\frac{t^{3\gamma }}{\Gamma (3\gamma +1)}. \end{array}$$

The LADM solution for Example 2 is
$$\nu \left(x,y,t\right)=\nu_{0}\left(x,y,t\right)+\nu_{1}\left(x,y,t\right)+\nu_{2}\left(x,y,t\right)+\nu_{3}\left(x,y,t\right)+...,$$
$$\begin{array}{c} \nu \left(x,y,t\right)=\cos\left(x+y\right)-2\sin\left(x+y\right)\frac{t^{\gamma }}{\Gamma (\gamma +1)}-4\cos\left(x+y\right)\frac{t^{2\gamma }}{\Gamma (2\gamma +1)}\\ +8\sin\left(x+y\right)\frac{t^{3\gamma }}{\Gamma (3\gamma +1)}+..., \end{array}$$
and so on. The solution in a series form is given by
(30)$$\begin{array}{c} \nu \left(x,y,t\right)=\cos\left(x+y\right)\left(1-\frac{4t^{2\gamma }}{\Gamma (2\gamma +1)}+\frac{16t^{2\gamma }}{\Gamma (2\gamma +1)}-\ldots \right)\\ -\sin\left(x+y\right)\left(\frac{2t^{\gamma }}{\Gamma (\gamma +1)}-\frac{8t^{3\gamma }}{\Gamma (3\gamma +1)}+\frac{32t^{5\gamma }}{\Gamma (5\gamma +1)}-\ldots \right), \end{array}$$
when γ=1, then LADM solution is
(31)$$\begin{array}{c} \nu (x,y,t)=\cos\left(x+y+2t\right). \end{array}$$

Figure 2 consists of four graphs; (a) the exact solution of ν(x,y,t) and (b) LADM solution ν(x,y,t) of Example 2 at γ=1. Figure 2a,b indicate that the present method has strong agreement with exact solution for the problem. In Figure 2c,d, two graphs (c) and (d) are given, that represents the analytical solution of Example 2 at fractional γ=0.75 and 0.50 respectively. Figure 2c,d reflects that as fractional order approaches to integer order the solution surfaces of fractional order are convergent to the integer order surface. It means that physically we can model any of the surfaces as desired by the physical phenomena occurring in nature.

**Example** **3.**
*Consider the following non-homogeneous fractional dispersive KdV equation [33]*
(32)$$\frac{\partial^{\gamma }\nu }{\partial t^{\gamma }}+\frac{\partial^{3}\nu }{\partial x^{3}}=-\sin\pi x\sin t-\pi^{3}\cos\pi x\cos t,\;0<\gamma \leq 1,$$
*with initial condition*
(33)ν(x,0)=sinπx.


Taking Laplace transform of Equation (Equation 32),
$$L\left[\frac{\partial^{\gamma }\nu }{\partial t^{\gamma }}\right]=L\left[-\sin\pi x\sin t-\pi^{3}\cos\pi x\cos t\right]-L\left[\frac{\partial^{3}\nu }{\partial x^{3}}\right],$$
$$s^{\gamma }L\left[\nu (x,t)\right]-s^{\gamma -1}\left[\nu (x,0)\right]=L\left[-\sin\pi x\sin t-\pi^{3}\cos\pi x\cos t\right]-L\left[\frac{\partial^{3}\nu }{\partial x^{3}}\right].$$

Applying an inverse Laplace transform
$$\nu (x,t)=L^{-1}\left[\frac{\nu (x,0)}{s}+\frac{1}{s^{\gamma }}L\left[-\sin\pi x\sin t-\pi^{3}\cos\pi x\cos t\right]-\frac{1}{s^{\gamma }}L\left[\frac{\partial^{3}\nu }{\partial x^{3}}\right]\right],$$
$$\begin{array}{c} \nu (x,t)=L^{-1}\left[\frac{\sin\pi x}{s}\right]+L^{-1}\left[\frac{1}{s^{\gamma }}L\left[-\sin\pi x\sin t-\pi^{3}\cos\pi x\cos t\right]\right]\\ -L^{-1}\left[\frac{1}{s^{\gamma }}L\left[\frac{\partial^{3}\nu }{\partial x^{3}}\right]\right]. \end{array}$$

Using the ADM procedure, we get
$$\begin{array}{c} \sum_{j=0}^{\infty }\nu_{j}\left(x,t\right)=L^{-1}\left[\frac{\sin\pi x}{s}\right]+L^{-1}\left[\frac{1}{s^{\gamma }}L\left[-\sin\pi x(t-\frac{t^{3}}{3!}+\frac{t^{5}}{5!}-\frac{t^{7}}{7!}+\frac{t^{9}}{9!})\right]\right]\\ +L^{-1}\left[\frac{1}{s^{\gamma }}L\left[-\pi^{3}\cos\pi x(1-\frac{t^{2}}{2!}+\frac{t^{4}}{4!}-\frac{t^{6}}{6!}+\frac{t^{8}}{8!})\right]\right]-L^{-1}\left[\frac{1}{s^{\gamma }}L\left[\sum_{j=0}^{\infty }\frac{\partial^{3}\nu_{j}}{\partial x^{3}}\right]\right], \end{array}$$
(34)$$\begin{array}{c} \nu_{0}\left(x,t\right)=\sin\pi x\\ -\sin\pi x\left(\frac{t^{\gamma +1}}{\Gamma (\gamma +2)}-\frac{t^{\gamma +3}}{\Gamma (\gamma +4)}+\frac{t^{\gamma +5}}{\Gamma (\gamma +6)}-\frac{t^{\gamma +7}}{\Gamma (\gamma +8)}+\frac{t^{\gamma +9}}{\Gamma (\gamma +10)}\right)\\ -\pi^{3}\cos\pi x\left(\frac{t^{\gamma }}{\Gamma (\gamma +1)}-\frac{t^{\gamma +2}}{\Gamma (\gamma +3)}+\frac{t^{\gamma +4}}{\Gamma (\gamma +5)}-\frac{t^{\gamma +6}}{\Gamma (\gamma +7)}+\frac{t^{\gamma +8}}{\Gamma (\gamma +9)}\right), \end{array}$$
$$\nu_{j+1}\left(x,t\right)=-L^{-1}\left[\frac{1}{s^{\gamma }}L\left[\sum_{j=0}^{\infty }\frac{\partial^{3}\nu_{j}}{\partial x^{3}}\right]\right],$$
for j=0,1,2,..
(35)$$\begin{array}{c} \nu_{1}\left(x,t\right)=-L^{-1}\left[\frac{1}{s^{\gamma }}L\left[\frac{\partial^{3}\nu_{0}}{\partial x^{3}}\right]\right],\\ \nu_{1}\left(x,t\right)=\pi^{3}\cos\pi x\frac{t^{\gamma }}{\Gamma (\gamma +1)}\\ -\pi^{3}\cos\pi x\left(\frac{t^{2\gamma +1}}{\Gamma (2\gamma +2)}-\frac{t^{2\gamma +3}}{\Gamma (2\gamma +4)}+\frac{t^{2\gamma +5}}{\Gamma (2\gamma +6)}-\frac{t^{2\gamma +7}}{\Gamma (2\gamma +8)}+\frac{t^{2\gamma +9}}{\Gamma (2\gamma +10)}\right)\\ +\pi^{6}\sin\pi x\left(\frac{t^{2\gamma }}{\Gamma (2\gamma +1)}-\frac{t^{2\gamma +2}}{\Gamma (2\gamma +3)}+\frac{t^{2\gamma +4}}{\Gamma (2\gamma +5)}-\frac{t^{2\gamma +6}}{\Gamma (2\gamma +7)}+\frac{t^{2\gamma +8}}{\Gamma (2\gamma +9)}\right),\\ \nu_{2}\left(x,t\right)=-L^{-1}\left[\frac{1}{s^{\gamma }}L\left[\frac{\partial^{3}\nu_{1}}{\partial x^{3}}\right]\right],\\ \nu_{2}\left(x,t\right)=-\pi^{6}\sin\pi x\frac{t^{2}\gamma }{\Gamma (2\gamma +1)}\\ +\pi^{6}\sin\pi x\left(\frac{t^{3\gamma +1}}{\Gamma (3\gamma +2)}-\frac{t^{3\gamma +3}}{\Gamma (3\gamma +4)}+\frac{t^{3\gamma +5}}{\Gamma (3\gamma +6)}-\frac{t^{3\gamma +7}}{\Gamma (3\gamma +8)}+\frac{t^{3\gamma +9}}{\Gamma (3\gamma +10)}\right)\\ +\pi^{9}\cos\pi x\left(\frac{t^{3\gamma }}{\Gamma (3\gamma +1)}-\frac{t^{3\gamma +2}}{\Gamma (3\gamma +3)}+\frac{t^{3\gamma +4}}{\Gamma (3\gamma +5)}-\frac{t^{3\gamma +6}}{\Gamma (3\gamma +7)}+\frac{t^{3\gamma +8}}{\Gamma (3\gamma +9)}\right). \end{array}$$

The LADM solution for Example 3 is
$$\nu \left(x,t\right)=\nu_{0}\left(x,t\right)+\nu_{1}\left(x,t\right)+\nu_{2}\left(x,t\right)+\nu_{3}\left(x,t\right)+\ldots ,$$
$$\begin{array}{c} \nu (x,t)=\sin\pi x\\ -\sin\pi x\left(\frac{t^{\gamma +1}}{\Gamma (\gamma +2)}-\frac{t^{\gamma +3}}{\Gamma (\gamma +4)}+\frac{t^{\gamma +5}}{\Gamma (\gamma +6)}-\frac{t^{\gamma +7}}{\Gamma (\gamma +8)}+\frac{t^{\gamma +9}}{\Gamma (\gamma +10)}\right)\\ -\pi^{3}\cos\pi x\left(\frac{t^{\gamma }}{\Gamma (\gamma +1)}-\frac{t^{\gamma +2}}{\Gamma (\gamma +3)}+\frac{t^{\gamma +4}}{\Gamma (\gamma +5)}-\frac{t^{\gamma +6}}{\Gamma (\gamma +7)}+\frac{t^{\gamma +8}}{\Gamma (\gamma +9)}\right)\\ +\pi^{3}\cos\pi x\frac{t^{\gamma }}{\Gamma (\gamma +1)}\\ -\pi^{3}\cos\pi x\left(\frac{t^{2\gamma +1}}{\Gamma (2\gamma +2)}-\frac{t^{2\gamma +3}}{\Gamma (2\gamma +4)}+\frac{t^{2\gamma +5}}{\Gamma (2\gamma +6)}-\frac{t^{2\gamma +7}}{\Gamma (2\gamma +8)}+\frac{t^{2\gamma +9}}{\Gamma (2\gamma +10)}\right)\\ +\pi^{6}\sin\pi x\left(\frac{t^{2\gamma }}{\Gamma (2\gamma +1)}-\frac{t^{2\gamma +2}}{\Gamma (2\gamma +3)}+\frac{t^{2\gamma +4}}{\Gamma (2\gamma +5)}-\frac{t^{2\gamma +6}}{\Gamma (2\gamma +7)}+\frac{t^{2\gamma +8}}{\Gamma (2\gamma +9)}\right)\\ -\pi^{6}\sin\pi x\frac{t^{2}\gamma }{\Gamma (2\gamma +1)}\\ +\pi^{6}\sin\pi x\left(\frac{t^{3\gamma +1}}{\Gamma (3\gamma +2)}-\frac{t^{3\gamma +3}}{\Gamma (3\gamma +4)}+\frac{t^{3\gamma +5}}{\Gamma (3\gamma +6)}-\frac{t^{3\gamma +7}}{\Gamma (3\gamma +8)}+\frac{t^{3\gamma +9}}{\Gamma (3\gamma +10)}\right)\\ +\pi^{9}\cos\pi x\left(\frac{t^{3\gamma }}{\Gamma (3\gamma +1)}-\frac{t^{3\gamma +2}}{\Gamma (3\gamma +3)}+\frac{t^{3\gamma +4}}{\Gamma (3\gamma +5)}-\frac{t^{3\gamma +6}}{\Gamma (3\gamma +7)}+\frac{t^{3\gamma +8}}{\Gamma (3\gamma +9)}\right)+... \end{array}$$
when γ=1, then the LADM solution is
(36)$$\begin{array}{c} \nu (x,t)=\sin\pi x\cos t. \end{array}$$

Figure 3 consists of three graphs; (a) the exact solution of ν(x,t) and (b) LADM solution ν(x,t) of Example 3 at γ=1. Figure 3a,b indicate that the present method has strong agreement with exact solution for the problem. In Figure 3c, graphs (c) are given, that represent the analytical solution of Example 3 at fractional γ=0.75 respectively. Figure 3c reflects that the fractional order approaches to integer order, the solution surfaces of fractional order are convergent to the integer order surface. It means that physically we can model, any of the surfaces as desired by the physical phenomena occurring in nature.

**Example** **4.**
*Consider the following non-homogeneous fractional dispersive KdV equation in three dimensional space [33]*
(37)$$\begin{array}{c} \frac{\partial^{\gamma }\nu }{\partial t^{\gamma }}+\frac{\partial^{3}\nu }{\partial x^{3}}+\frac{1}{8}\frac{\partial^{3}\nu }{\partial y^{3}}+\frac{1}{27}\frac{\partial^{3}\nu }{\partial z^{3}}=-\sin(x+2y+3z)\cos t\\ +\sin(x+2y+3z)\cos t,\;t>0,\;0<\gamma \leq 1, \end{array}$$
*with the initial condition*
(38)ν(x,y,z,0)=0.


Taking Laplace transform of Equation (Equation 37),
$$\begin{array}{c} L\left[\frac{\partial^{\gamma }\nu }{\partial t^{\gamma }}\right]=L\left[\sin(x+2y+3z)\cos t\right]-L\left[3\cos(x+2y+3z)\sin t\right]\\ -L\left[\frac{\partial^{3}\nu }{\partial x^{3}}+\frac{1}{8}\frac{\partial^{3}\nu }{\partial y^{3}}+\frac{1}{27}\frac{\partial^{3}\nu }{\partial z^{3}}\right], \end{array}$$
$$\begin{array}{c} s^{\gamma }L\left[\nu (x,y,z,t)\right]-s^{\gamma -1}\left[\nu (x,y,z,0)\right]=L\left[\sin\left(x+2y+3z\right)\cos t\right]\\ -L\left[3\cos(x+2y+3z)\sin t\right]-L\left[\frac{\partial^{3}\nu }{\partial x^{3}}+\frac{1}{8}\frac{\partial^{3}\nu }{\partial y^{3}}+\frac{1}{27}\frac{\partial^{3}\nu }{\partial z^{3}}\right]. \end{array}$$

Applying inverse Laplace transform
$$\begin{array}{c} \nu (x,y,z,t)=L^{-1}\left[\frac{\nu (x,y,z,0)}{s}+\frac{1}{s^{\gamma }}L\left[\sin\left(x+2y+3z\right)\cos t\right]\right]\\ +L^{-1}\left[\frac{1}{s^{\gamma }}L\left[-3\cos\left(x+2y+3z\right)\sin t\right]\right]-L^{-1}\left[\frac{1}{s^{\gamma }}L\left[\frac{\partial^{3}\nu }{\partial x^{3}}+\frac{1}{8}\frac{\partial^{3}\nu }{\partial y^{3}}+\frac{1}{27}\frac{\partial^{3}\nu }{\partial z^{3}}\right]\right], \end{array}$$
$$\nu_{0}\left(x,y,z,t\right)=L^{-1}\left[\frac{1}{s^{\gamma }}L\left[\sin\left(x+2y+3z\right)\left(1-\frac{t^{2}}{2!}+\frac{t^{4}}{4!}-\frac{t^{6}}{6!}+\frac{t^{8}}{8!}\right)\right]\right].$$

Using the ADM procedure, we get
$$\begin{array}{c} \sum_{j=0}^{\infty }\nu_{j}\left(x,y,t\right)=L^{-1}\left[\frac{1}{s^{\gamma }}L\left[-3\cos\left(x+2y+3z\right)\left(t-\frac{t^{3}}{3!}+\frac{t^{5}}{5!}-\frac{t^{7}}{7!}+\frac{t^{9}}{9!}\right)\right]\right]\\ -L^{-1}\left[\frac{1}{s^{\gamma }}L\left[\sum_{j=0}^{\infty }\frac{\partial^{3}\nu_{j}}{\partial x^{3}}+\frac{1}{8}\sum_{j=0}^{\infty }\frac{\partial^{3}\nu_{j}}{\partial y^{3}}+\frac{1}{27}\sum_{j=0}^{\infty }\frac{\partial^{3}\nu_{j}}{\partial z^{3}}\right]\right], \end{array}$$
(39)$$\begin{array}{c} \nu_{0}\left(x,y,z,t\right)=\\ \sin\left(x+2y+3z\right)\left(\frac{t^{\gamma }}{\Gamma (\gamma +1)}-\frac{t^{\gamma +2}}{\Gamma (\gamma +3)}+\frac{t^{\gamma +4}}{\Gamma (\gamma +5)}-\frac{t^{\gamma +6}}{\Gamma (\gamma +7)}+\frac{t^{\gamma +8}}{\Gamma (\gamma +9)}\right), \end{array}$$
$$\begin{array}{c} \nu_{1}\left(x,y,z,t\right)=L^{-1}\left[\frac{1}{s^{\gamma }}L\left[-3\cos\left(x+2y+3z\right)\left(t-\frac{t^{3}}{3!}+\frac{t^{5}}{5!}-\frac{t^{7}}{7!}+\frac{t^{9}}{9!}\right)\right]\right]\\ -L^{-1}\left[\frac{1}{s^{\gamma }}L\left[\frac{\partial^{3}\nu_{0}}{\partial x^{3}}+\frac{1}{8}\frac{\partial^{3}\nu_{0}}{\partial y^{3}}+\frac{1}{27}\frac{\partial^{3}\nu_{0}}{\partial z^{3}}\right]\right], \end{array}$$
$$\nu_{j+1}\left(x,y,z,t\right)=-L^{-1}\left[\frac{1}{s^{\gamma }}L\left[\sum_{j=0}^{\infty }\frac{\partial^{3}\nu_{j}}{\partial x^{3}}+\frac{1}{8}\sum_{j=0}^{\infty }\frac{\partial^{3}\nu_{j}}{\partial y^{3}}+\frac{1}{27}\sum_{j=0}^{\infty }\frac{\partial^{3}\nu_{j}}{\partial z^{3}}\right]\right],$$
for j=0,1,2,…
(40)$$\begin{array}{c} \nu_{1}\left(x,y,z,t\right)=0,\\ \nu_{j+1}\left(x,y,z,t\right)=0. \end{array}$$

This readily yields the exact solution
(41)$$\begin{array}{c} \nu (x,y,z,t)=\\ \sin\left(x+2y+3z\right)\left(\frac{t^{\gamma }}{\Gamma (\gamma +1)}-\frac{t^{\gamma +2}}{\Gamma (\gamma +3)}+\frac{t^{\gamma +4}}{\Gamma (\gamma +5)}-\frac{t^{\gamma +6}}{\Gamma (\gamma +7)}+\frac{t^{\gamma +8}}{\Gamma (\gamma +9)}\right), \end{array}$$
when γ=1, then LADM solution is
(42)$$\begin{array}{c} \nu (x,y,t)=\sin\left(x+2y+3z\right)\sin t. \end{array}$$

Figure 4 consists of two graphs; (a) the exact solution of ν(x,y,t) and (b) the LADM solution ν(x,y,t) of Example 4 at γ=1. Figure 4a,b indicate that the present method has strong agreement with exact solution for the problem. In Figure 4c,d, two graphs (c) and (d) are given, that represent the analytical solution of Example 4 at fractional γ=0.75 and 0.50 respectively. Figure 4c,d reflects that, as fractional order approaches to integer order, the solution surfaces of fractional order are convergent to the integer order surface. It means that physically we can model, any of the surfaces as desired by the physical phenomena occurring in nature.

## 5. Conclusions

In this paper, the analytical solutions of third order dispersive fractional partial differential equations are determined, using LADM. The fractional derivatives are described by the Caputo operator. The LADM, solutions are obtained at fractional and integer orders for all problems. The results revealed the highest agreement with the exact solutions for the problems. The LADM solutions for some numerical examples have shown the validity of the proposed method. It is also investigated that the fractional order solutions are convergent to the exact solution for the problems as fractional order approaches to integer order. The implementation of LADM to illustrative examples have also confirmed that the fractional order mathematical model can be the best representation of any experimental data as compare to integer order model. Moreover, by taking different fractional orders, we can find a way to set suitable mathematical model for any experimental data, and thus found reasonable consequences. Hence, it is concluded that LADM is the best tool for the solution of FPDEs, as compare to ADM, VIM and DTM in literature. LADM provide the highest rate of convergence to the exact solution for the problems. In future, LADM can be used to find the analytical solution of other non-linear FPDEs, which are frequently used in science and engineering. LADM, solutions for fractional order problems will prove the better understanding of the real world problems represented by FPDEs.

## Figures and Tables

**Figure 1 entropy-21-00335-f001:**
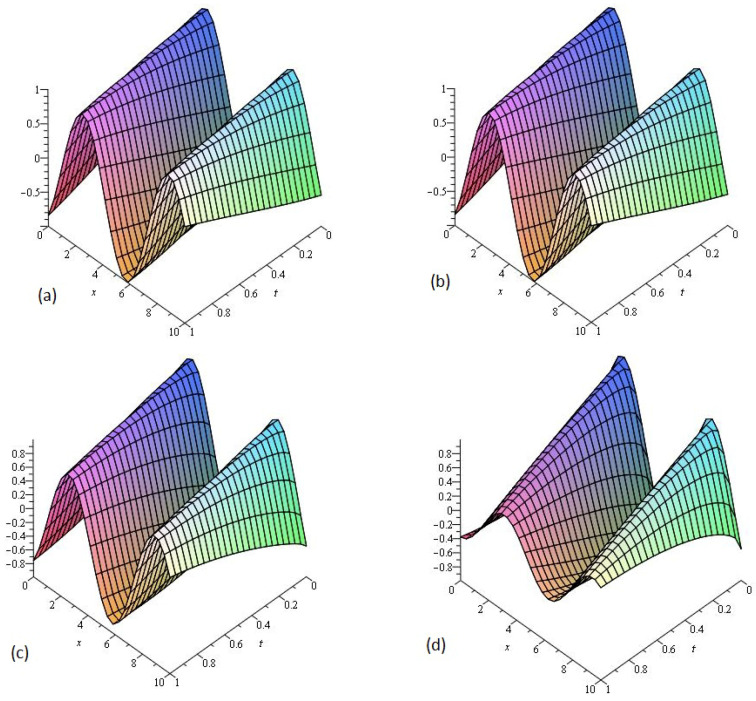
The (**a**) exact and (**b**) Laplace–Adomian decomposition method (LADM) solutions of ν(x,t) of Example 1, at γ=1. The LADM solution of ν(x,t) of Example 1, at (**c**) γ=0.75 and (**d**) γ=0.50.

**Figure 2 entropy-21-00335-f002:**
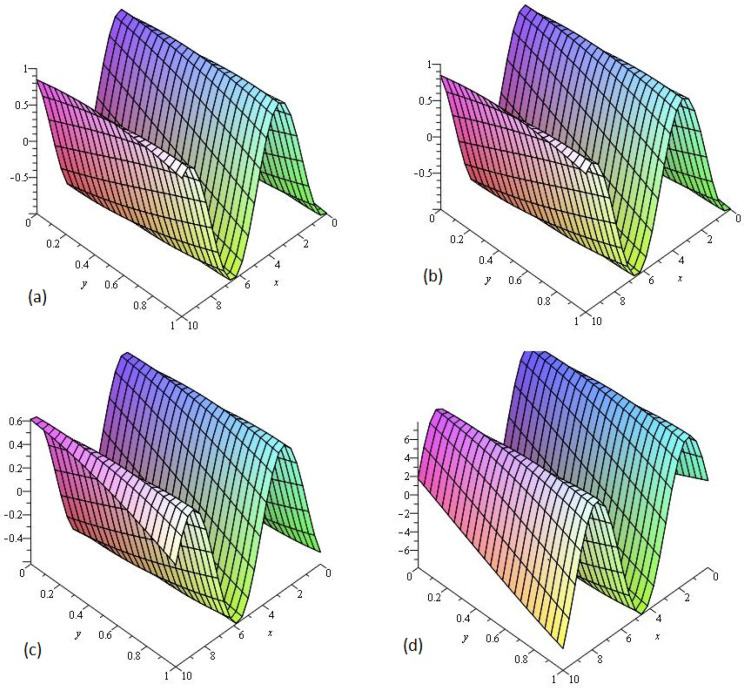
The (**a**) Exact and (**b**) LADM solutions of ν(x,y,t) of Example 2, at γ=1. The LADM solution of ν(x,y,t) of Example 2, at (**c**) γ=0.75 and (**d**) γ=0.50.

**Figure 3 entropy-21-00335-f003:**
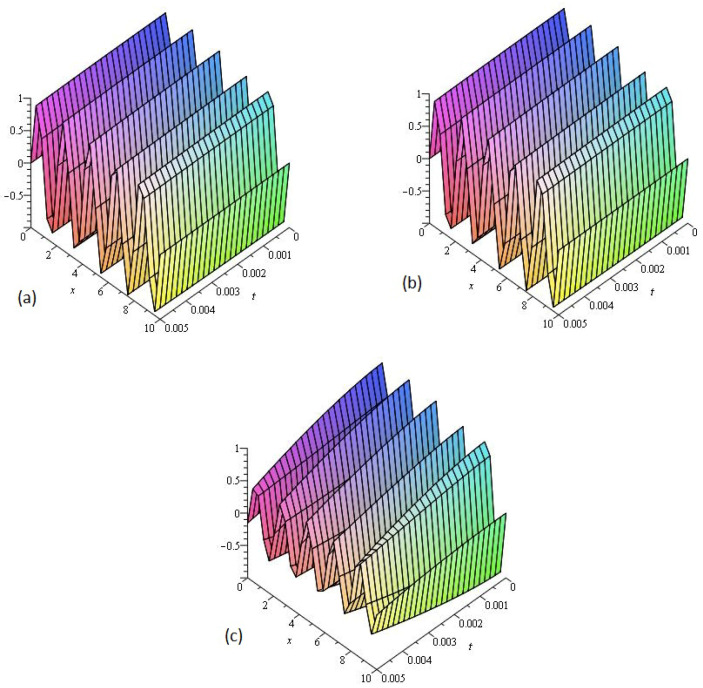
The (**a**) Exact and (**b**) LADM solutions of ν(x,t) of Example 3, at γ=1. The LADM solution of ν(x,t) of Example 3, at (**c**) γ=0.75.

**Figure 4 entropy-21-00335-f004:**
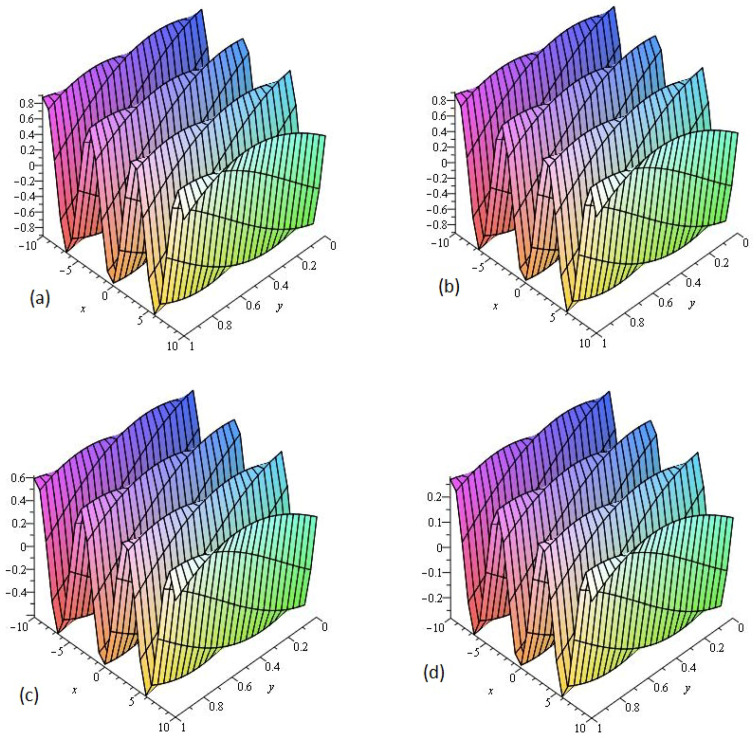
The (**a**) exact and (**b**) LADM solutions of ν(x,y,z,t) of Example 4, at γ=1. The LADM solution of ν(x,y,z,t) of Example 4, at (**c**) γ=0.75 and (**d**) γ=0.50.

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
