# Peer review of "Application of Laplace–Adomian Decomposition Method for the Analytical Solution of Third-Order Dispersive Fractional Partial Differential Equations"

_entropy, 2019, doi:10.3390/e21040335_

Reviewer 1 Report

Dear authors,

The Results section should be improved, despite the Figures show the effectiveness of the proposed method it is desirable that all Figures be referenced in the text and discussed.

Improve the conclusion section based on the results section.

Author Response

I would like to thanks to the reviewer for giving good suggestions to improve the current status of this manuscript.

Point 1: The Results section should be improved, despite the Figures show the effectiveness of the proposed method it is desirable that all Figures be referenced in the text and discussed.

Response 1: The result section is improved by adding discussion to all figures in the text. For reference see page no. 8, 9, 13, 16 and line numbers 79-85, 87-93, 95-101,103-109.

Point 2: Improve the conclusion section based on the results section.

Response 2: The conclusion section is improved on basis of results section. See page no. 16 and line numbers 117-123.

Reviewer 2 Report

In my opinion, the paper presents some details. A MINOR revision is required to make the manuscript worth publishing.

It is proper to add some convergence theories to support the numerical results, if it is possible. In addition, an overall review may be needed for fixing the grammatical errors in the manuscript.

What is the effectiveness the proposed method over the existing method? Comparison should be done with the latest reported method in (2016/2017/2018) papers.

I recommend a revision of the introduction. In my opinion, you should explain your results in more detail in the introduction

Check the format of the journal and made all the references according to the journal style.

Please add the following references

Fractional Li\'enard type model of a pipeline within the fractional derivative without singular kernel. Advances in Difference Equations, 2016(1), 2016, 1-17.

On the solutions of fractional order of evolution equations. The European Physical Journal Plus, 132(1), 2017, 1-17.

The Feng's first integral method applied to the nonlinear mKdV space-time fractional partial differential equation. Rev. Mex. F\'is, 62(4), 2016, 310-316.

Future research direction will be shown in Conclusion.

I want to read speedly the last version of paper before publishing if it possible for you.

Author Response

I am very pleased with the suggestions of the reviewer. He gave very good opinions for the improvement of this manuscript. 

Point 1: It is proper to add some convergence theories to support the numerical results, if it is possible. In addition, an overall review may be needed for fixing the grammatical errors in the manuscript. 

Response 1: The convergence theorem is included in the manuscript on page no. 4 and line numbers 71-73.

Also we did the overall review of the manuscript grammatically and correct the spelling and grammatical errors as

Ø  Page no.1, line 3 (replace defined by define), line 5 (replace conform by confirming), line 22 (have is replaced by has) line 23 (higher order is replaced by higher-order).

Ø   Page no. 2, line 32 (laplace is replaced by Laplace), line 42 (Pre-defined is replaced by predefined), line 43 (considered as an ideal is replaced by considered an ideal).

Ø   Page no. 16, line 115 (higher is replaced by highest), line 115 (of is replaced by for), line 117 (of is replaced by for)

Point 2: What is the effectiveness the proposed method over the existing method? Comparison should be done with the latest reported method in (2016/2017/2018) papers.

Response 2: We made the comparison with the methods by using references on page no. 2 and line 34-38. Also see reference no. 16-21.  

Point 3:  I recommend a revision of the introduction. In my opinion, you should explain your results in more detail in the introduction

Response 3:  The introduction of the manuscript is revised by adding some important references related to this research paper. See page no. 2, line no. 34-38.  

Point 4: Check the format of the journal and made all the references according to the journal style.

Response 4: We, did it as suggested

Point 5: Please add the following references

Fractional Li\'enard type model of a pipeline within the fractional derivative without singular kernel. Advances in Difference Equations, 2016(1), 2016, 1-17.

On the solutions of fractional order of evolution equations. The European Physical Journal Plus, 132(1), 2017, 1-17.

The Feng's first integral method applied to the nonlinear mKdV space-time fractional partial differential equation. Rev. Mex. F\'is, 62(4), 2016, 310-316

Response 5: We cite the above references mentioned in point 5, on page 2, line 35, page 3, lines 67-68. In bibliography, its reference numbers are 16, 32 and 33.  

Point 6: Future research direction will be shown in Conclusion.

Response 6: Future work is included in the conclusion of the manuscript on page no. 16, line (123-126).

Reviewer 3 Report

This paper provided the Laplace-Adomian decomposition method to obtain the analytical solutions of fractional differential equations. Some numerical examples are considered. 

Some remarks:

There are many English typing errors, case sensitivity, RK4, FPDE....

In definition 2.1, it should be say: the fractional integral of Riemann-Liouville type or R-L fractional integral

The definition 2.2 is not right.

The first and third formula in Lemma 2.3 should be correct。

Generally, the star * is used for the convolution, and the two formula in Def. 2.5 is not right.

u(x_1,t_1) should be changed to u(x,t).

More important, the numerical results should be compared with results in literature. Otherwise, how to determine the correctness of the numerical results in this manuscript.

Author Response

I am very thankful to the reviewer for giving me valuable suggestions.

Round  2

Reviewer 3 Report

This version can be accepted. 

Remarks: The title in section 4 is Example or Eample?